# Effect of Solid Fat Content in Fat Droplets on Creamy Mouthfeel of Acid Milk Gels

**DOI:** 10.3390/foods11192932

**Published:** 2022-09-20

**Authors:** Hui Zhou, Yan Zhao, Di Fan, Qingwu Shen, Chengguo Liu, Jie Luo

**Affiliations:** 1College of Food Science and Technology, Hunan Agricultural University, Changsha 410114, China; 2Key Laboratory of Functional Dairy, College of Food Science and Nutritional Engineering, China Agricultural University, Beijing 100083, China

**Keywords:** solid fat content, acid milk gel, creamy mouthfeel, tribology, in-mouth coalescence

## Abstract

Previous studies have shown that emulsions with higher solid fat content (SFC) are related to a higher in-mouth coalescence level and fat-related perception. However, the effect of SFC in fat droplets on the fat-related attributes of emulsion-filled gels has not been fully elucidated. In this study, the effect of SFC on the creamy mouthfeel of acid milk gel was investigated. Five kinds of blended milk fats with SFC values ranging from 10.61% to 85.87% were prepared. All crystals in the blended milk fats were needle-like, but the onset melting temperature varied widely. Blended milk fats were then mixed with skim milk to prepare acid milk gels (EG10–EG85, fat content 3.0%). After simulated oral processing, the particle size distribution and confocal images of the gel bolus showed that the degree of droplet coalescence in descending order was EG40 > EG20 > EG60 > EG10 ≥ EG85. There was no significant difference in apparent viscosity measured at a shear rate of 50/s between bolus gels, but the friction coefficients measured at 20 mm/s by a tribological method were negatively correlated with the coalescence result. Furthermore, quantitative descriptive analysis and temporal dominance of sensations analysis showed that SFC significantly affected the ratings of melting, mouth coating, smoothness and overall creaminess, as well as the perceived sequence and the duration of melting, smoothness and mouth coating of acid milk gels. Overall, our study highlights the role of intermediate SFC in fat droplets on the creamy mouthfeel of acid milk gels, which may contribute to the development of low-fat foods with desirable sensory perception.

## 1. Introduction

Creaminess is an important sensory attribute of dairy products and high-fat nondairy foods, commonly reflecting pleasant and satisfying sensory characteristics [1]. The amount of fat present in foods has been found to play a critical role in the perception of creaminess [2]. However, with a heightened awareness of fat content and the demand for low-fat products, enhancing the creaminess perception without increasing or even reducing the fat content of foods has received great attention from researchers and the food industry.

Modulating droplet instability against in-mouth coalescence has been found to be an effective method to enhance the sensation of creamy mouthfeel [3]. During oral processing, emulsion-based foods are affected by various oral environmental conditions [4], inducing changes in sensory perception [5]. Partial coalescence is a major form of oral destabilization in many emulsion-based foods in which emulsion droplets are partially crystallized [6]. Partial coalescence takes place when crystals formed inside the fat droplet pierce the interface and enter the triglyceride core region of another droplet, and with increasing time, the two particles fuse more closely to reduce the total surface area exposed to the aqueous phase [7]. Fat crystals that promote partial coalescence melt in the oral cavity, resulting in an increased size of fat droplets and the enhancement of creamy mouthfeel [3]. Therefore, promoting the partial coalescence of fat droplets in emulsion-based foods can be a strategy to increase in-mouth coalescence and thus creamy mouthfeel.

The size, number and orientation of crystals in the interface and the properties of the adsorption layer at the droplet interface affect the stability and the final coalescence of crystal-damaged films [8]. Previous studies have shown that emulsions with lower solid fat content (SFC) values coalesced less than emulsions with higher SFC values under shear in an optical tribometer [3,9]. More importantly, emulsions with higher SFC can evoke a stronger fat-related perception to a certain extent [3,6]. However, previous study [9] demonstrated that, in emulsion-filled gels, an increase in SFC decreased friction but did not affect the perception of fat-related attributes. They deduced that differences in the breakdown behavior of gels and emulsions led to different perceptions of solid fat [9]. In contrast to emulsions, emulsion droplets in emulsion-filled gels coalesce in a biopolymer matrix, so the properties of emulsion gels are affected by the properties of both the gel matrix and the emulsion droplets [5]. Furthermore, the value of SFC might not be high enough such that it could disappear at an early stage of oral processing before partial coalescence occurs [3]. However, an excessively high SFC value will trap the remaining liquid fat in the pores of the crystal network and also hinder coalescence [10]. So far, it has not been clear how much SFC in emulsion-filled gel promotes the in-mouth coalescence of fat droplets and whether SFC affects the creamy mouthfeel of emulsion-filled gels.

During oral processing, the matrix of an emulsion-filled gel is first broken down by shearing and melting, followed by the release of fat droplets from the gel matrix and then the coalescence of fat droplets. Therefore, the perception of an emulsion-filled gel is highly dynamic. Temporal dominance of sensations (TDS) is a methodology that complements static multidimensional sensory profiling by providing a way to simultaneously assess several attributes dynamically over time. Compared with descriptive analysis (QDA), TDS is a much simpler task, feasible for untrained consumers, and is currently extensively used in food industries [11].

Acid milk gel is a typical emulsion-filled gel. In this study, the effect of SFC in fat droplets on the creamy mouthfeel of acid milk gel was investigated. Five kinds of blended milk fats with SFC values ranging from 10% to 85% were prepared. The SFC, crystal morphology and thermodynamic properties of the blended milk fats were characterized. Then, acid milk gels differing in SFC values were prepared, and the microstructure, apparent viscosity and tribological properties of acid milk gels were evaluated after in vitro simulated oral processing. The creamy mouthfeel evaluation of acid milk gels was performed using both QDA and TDS analysis.

## 2. Materials and Methods

### 2.1. Materials

Anhydrous milk fat (fat content 99.9%, Friesland Campina, The Netherlands), skimmed milk powder (Anchor, Auckland, New Zealand) and skim milk (Yili Ltd., Inner Mongolia, China) were used. Chemical-grade oleic acid glyceride (CP, 60%) was purchased from Shanghai Macklin Biochemical Company, China, while glucono-δ-lactone (GDL), chemical-grade glycerin stearate (99.8%) and sodium dodecyl were purchased from Sigma–Aldrich (St. Louis, MO, USA). Food-grade glyceride stearate and oleic acid glyceride were obtained from Shi Hua Shi Shuo Raw Material Shop (Hebei, China).

Salivette (SARSTEDT, Dachau, Germany) was used to collect human saliva [12]. Five males and five females (23–26 years old, healthy adults) were selected. After saliva collection, all of the tubes were centrifuged (1000× *g*, 5 min), and then the centrifuged saliva was mixed thoroughly and stored at −80 °C.

### 2.2. Blended Milk Fats with Different Solid Fat Content Values

#### 2.2.1. Preparation of the Blended Milk Fats

Based on anhydrous milk fat, certain amounts of stearic acid glyceride and oleic acid glyceride were used to formulate blended milk fats with 10–85% SFC, as shown in Table 1. The blended milk fats were named F10, F20, F40, F60 and F85.

#### 2.2.2. Determination of Solid Fat Content

The proportion of SFC in blended milk fats (F10–F85) was determined by using nuclear magnetic resonance (P-NMR, PQ001-SFC, Shanghai, China) as described in the Chinese standard (GB/T 31743-2015) [13]. Three standard silicone oil samples with SFCs of 0%, 30% and 70% were used for calibration. The milk fat samples were prepared by maintaining them at 80 °C for 30 min to eliminate the historical crystallization of samples in P-NMR and then crystallized at 4 °C for 12 h. The blended milk fat samples were analyzed.

#### 2.2.3. Crystal Morphology

The microstructure of blended milk fat samples was analyzed using polarized light microscopy (PLM, OLYMPUS DP27, Tokyo, Japan) as described in a previous study [14] with some modifications. In order to erase crystal memory, blended milk fats were preheated at 80 °C for 30 min. The melted fat (5 µL) was placed on a preheated microscope slide with a preheated capillary tube and covered with a preheated cover. After that, the samples were stored in a refrigerator at 4 °C. After 12 h, the samples were imaged using PLM at a temperature of 4 °C, and all images were taken with a 50× objective.

#### 2.2.4. Thermodynamic Properties

Differential scanning calorimetry (DSC) (Mettler-Toledo, Zurich, Switzerland) with mechanical refrigeration was used to analyze the nonisothermal crystallization and melting thermodynamic properties of the samples [15]. Approximately 5–15 mg of each milk fat blend was loaded into hermetically sealed aluminum pans. The samples were first heated at 70 °C for 30 min and then cooled to 4 °C (5 °C/min) and maintained for 5 min, and then the temperature was increased to 70 °C (5 °C/min). The calorimetric parameters retrieved were those of crystallization and melting.

The STARe Excellence software (METTLER TOLEDO, Zurich, Switzerland) was used to obtain the crystallization temperature (T_C_, °C), the melting temperatures (T_M_, °C) and the total variation in enthalpy (Δ_C,Total_, Δ_M,Total_, J g^−1^).

### 2.3. Preparation and Characterization of Acid Milk Gels

#### 2.3.1. Preparation of Acid Milk Gels

Five kinds of blended milk fats (F10–F85) were used as the oil phase in a 70 °C water bath for 30 min to melt completely. A Peltier system was used for temperature control. Skimmed milk powder and Milli-Q water were reconstituted into skimmed milk at a ratio of 1:9 (*w*/*w*). The skimmed milk was stirred (1000 rpm, 10 min, 25 °C) with a magnetic stirrer to fully dissolve, and it was refrigerated overnight at 4 °C. The melted blended milk fats were then separately mixed with skimmed milk at a ratio of 3.0% (*w*/*v*) at 60 °C and emulsified using a T10 high-speed shearing apparatus (T10-basic, IKA, Königswinter, Germany) at 15,000 rpm with different shear times (5.5 min, 5.5 min, 6 min, 6.5 min and 6.5 min) to prepare milk emulsions with similar particle sizes, named E10, E20, E40, E60 and E85.

The acid-induced milk gels were prepared by adding 1.1% (*w*/*v*) GDL separately to milk emulsions E10–E85 and kept at 42 °C. When the pH value of the gel system dropped to 4.60 ± 0.02, the gels were kept at 4 °C overnight [16]. The gels were named EG10, EG20, EG40, EG60 and EG85. The apparent viscosity of acid milk gel samples was measured with a rheometer (Type AR2000, TA Instruments, New Castle, DE, USA) at a 1000 µm gap using a 40 mm stainless steel parallel plate probe at 4 °C [17]. The particle size distribution of fat droplets in the acid milk gel was determined by using a laser diffraction particle size analyzer (Type Mastersizer 3000, Malvern, UK) [18]. Casein micelles were dissociated by adding 1% sodium dodecyl sulfate to the gel. The volume diameters D_1_[4,3] of fat droplets were calculated. The apparent viscosity at a shear rate of 50/s was measured to represent the oral viscosity [19].

#### 2.3.2. Simulated Oral Processing

Acid milk gel samples were mixed with saliva at a ratio of 1:10 (*w*/*w*) and stirred manually at a shear rate of around 50 rpm at 37 °C for 20 s to simulate oral chewing [20], with slight modifications in the chewing time. In our sensory evaluation, panelists were instructed to chew 20 g of sample for 20 s; therefore, the simulated oral processing time was set at 20 s. After simulated oral processing, the semi-solid gel became a more liquid gel bolus, and the gel bolus was subjected to further characterization.

#### 2.3.3. Characterization of Gel Bolus

##### Particle Size Distribution

The particle sizes of fat droplets in the gel bolus were determined as described above. The volume diameters D_2_[4,3] were calculated.

##### Observation of the Microstructure

The coalescence of fat droplets in the gel bolus was observed by using a laser confocal microscope (Nikon Ltd., Tokyo, Japan) [21]. The fat and protein in samples were separately stained with Nile red and Fast green (both 1 mg/mL) with light avoidance for more than 2 h after mixing. A dyed sample (10 μL) was observed. All images were taken with a 100× objective lens with fluorescence excitation and emission wavelengths of 633 nm and 560 nm, respectively.

##### Apparent Viscosity

The apparent viscosities of the gel bolus were determined as described above but measured at 37 °C.

##### Tribological Properties

The friction properties of the gel bolus were determined using a stress-controlled rheometer (MCR301, Anton Paar, Austria) [12]. Friction coefficients were measured on a ball-on-three-pins setup with a spherical stainless steel ball (diameter: 12.7 mm), and polydimethyl siloxane (PDMS) films (4 mm × 5.9 mm × 15.9 mm) were used to simulate the oral surface. The gel bolus (2.00 ± 0.02 g) was put between the stainless steel ball and the plates. The normal force (6.5 N) was evenly distributed during measurements. The sliding speed was set at 0.1–500 mm/s, and the measurements were made at 37 °C.

#### 2.3.4. Sensory Evaluation of Acid Milk Gels

##### Quantitative Descriptive Analysis

QDA was performed by a trained panel consisting of four males and six females with an age range of 23–26 years. The creamy mouthfeel sensory attribute list and evaluation criteria are shown in Table 2 and were measured using a 10-point categorical scale according to a previous study [22]. Each sensory assessor was trained 4 times (each lasting approximately 2 h) for 2 weeks, which allowed the assessors to consistently identify and score attributes (graininess, melting, spreadability, thickness, smoothness, mouth coating and creaminess). Acid milk gels (20 g each) were served in 100 mL odorless transparent plastic cups coded with 3-digit random numbers. Participants were instructed to chew samples for 20 s and then swallow. Each trial consisted of 1 sample, with 2 min intervals between trials. The panelists were asked to rinse their mouths with water during the intervals. The acid milk gels were served to the panelists at 4 °C by placing them in an ice box during sensory evaluation to simulate normal consumption temperature in day-to-day scenarios.

##### Temporal Dominance of Sensations Analysis

TDS analysis was performed with a panel consisting of 28 participants (age range 23–26 years). Each participant was trained twice with a total of 6 h of training to be familiar with the definitions of sensory attributes and the procedure. The TDS analysis of creamy mouthfeel included 6 attributes: graininess, melting, spreadability, thickness, mouth coating and smoothness. The sensory attribute list and evaluation criteria are shown in Table 2. According to the chewing results of acid milk gels in pre-experiments, 20 s was sufficient time to complete the oral processing process, while swallowing and then aftertaste could last about 10 s. Therefore, the whole evaluation period was 30 s in our study. The test was conducted using Excel software (Microsoft Corporation, Seattle, WA, USA) with a timer developed by the lab. All TDS attributes were displayed on a computer screen. Participants selected the dominant attributes that attracted their attention at a specific time point. For each sample and bite, the participants placed a scoop of acid milk gel into their mouths and clicked the “START” button at the same time. After that, participants selected the dominant attributes at each time interval (5 s). After swallowing at approximately 20 s, the participants still evaluated the dominant sensations at each time interval (5 s) until the ending time (30 s). Participants could select more than one dominant attribute, select the same attribute multiple times, or not select any attribute at a certain time. The TDS curves were plotted with two additional lines for the chance and significance levels according to the formula described in previous study [23].

### 2.4. Statistical Analysis

All analytical measurements were performed in triplicate. The data were analyzed by multiway ANOVA of variance (SPSS 22.0, SPSS Inc., Chicago, IL, USA) with Duncan’s multiple range test. The statistically significant difference was set at *p* < 0.05. When testing the effect of SFC on the sensory properties, the fixed factor was samples, and the random factor was participants.

## 3. Results and Discussion

### 3.1. Crystallization Behavior of Blended Milk Fats

#### 3.1.1. Solid Fat Content

As shown in Table 3, there were significant differences in the SFC in different blended milk fats (*p* < 0.05). The SFC values in five blended milk fats in ascending order were F10 < F20 < F40 < F60 < F85, ranging from 10.61% to 85.87%, basically consistent with the set range (Table 1). Therefore, blended milk fats with different SFCs were successfully prepared. The fatty acid composition of five blended milk fats (F10–F85) is shown in the Appendix A, further confirming the increase in SFC in blended milk fats F10–F85.

#### 3.1.2. Crystal Morphology

The microscopic morphology of crystals in blended milk fats is shown in Figure 1. All crystals in the blended milk fats were needle-like (N-type crystals), which is consistent with the observations [24]. The needle-like structure of the crystal is able to penetrate the interface of fat droplets and cause crystal bridging of the fat droplets [25]. In contrast, the number of crystals in different blended milk fats was noticeably different. With the increase in SFC in the blended milk fats, the network structure of fat crystals became denser, and the number of crystals increased (Figure 1). Therefore, the influence of crystal shape on the partial coalescence of fat droplets and the mouthfeel of acid milk gels can be eliminated in our study.

#### 3.1.3. Thermodynamic Properties

Figure 2A,B shows the nonisothermal crystallization curves and melting curves of blended milk fats. During the whole crystallization process, F10, F20 and F85 had one crystallization peak, while F40 and F60 had two crystallization peaks, probably due to their broader fatty acid composition (Appendix A). With the increase in SFC in the blended milk fats, the onset temperature of crystallization increased, ranging from 7.79 ± 0.04 °C (F10) to 42.78 ± 0.08 °C (F85) (Figure 2C). The total exothermic enthalpy of blended milk fat also increased with the SFC value. In the melting process, there was only one melting peak for F10 and F85, two melting peaks for F20 and three melting peaks for F40 and F60. Different melting peaks correspond to different melting triglycerides in milk fats [26]. In addition, the onset melting temperature of blended milk fats significantly increased with the SFC in the blended milk fats (*p* < 0.05). F10 began to melt at 7.38 ± 0.08 °C, followed by F20 at 9.16 ± 0.04 °C. The onset melting temperature of F85 reached 57.45 ± 0.06 °C (Figure 2C), which was attributed to the fact that it had the highest saturated fatty acid content, which also contributed to the highest total endothermic enthalpy value.

During oral processing, the oral cavity can be either heated up or cooled down depending on the temperature of the consumed food, which in turn physically affects the perception of the texture of the food [27]. For acid milk gels that are usually consumed at low temperatures, the solid fat in the fat droplets will undergo a solid-to-liquid phase transition at the in-mouth physiological temperature of 37 °C. In our study, the onset melting temperature of F10–F85 ranged from approximately 7 °C to 58 °C. Therefore, at the oral temperature, the blended milk fat F10 would be expected to melt completely, while F85 might exist mainly in crystalline form, and F20, F40 and F60 would melt to varying degrees. The melting degree would exert a great effect on the mouthfeel attributes “melting” and “creaminess”, which might be related to the lubrication effect of the molten fat layer in the oral cavity [28].

### 3.2. The Partial Coalescence of Fat Droplets in Gel Bolus

#### 3.2.1. The Particle Size of Fat Droplets

The prepared emulsion samples had similar particle size distributions, in which the fat droplet diameter was about 4 μm, basically consistent with the size distribution of fat droplets in milk (the results are not shown). After gelation, the fat droplets in the acid milk gels coalesced to a certain extent, although the degree of coalescence was limited (Figure 3A). During the gelation process, the cross-linking of casein micelles will bring the fat droplets in the network closer, which may contribute to the aggregation and coalescence of fat droplets [21,29]. The volume diameters D_1_[4,3] of fat droplets in EG20 and EG40 were significantly higher than those in other gels (Figure 3C, *p* < 0.05). After simulated oral processing, the particle size distribution curves of gels moved significantly toward the right (Figure 3B), indicating the coalescence of fat droplets in the acid milk gels under simulated oral processing. The degree of coalescence of the EG40 bolus was the highest, with the D_2_[4,3] diameter increasing to 27.50 ± 0.01 µm (Figure 3C), followed by EG20 (12.06 ± 0.03 µm) and EG60 (7.92 ± 0.00 µm). There was no significant difference between the gel boluses of EG10 and EG85 (*p* > 0.05).

#### 3.2.2. Confocal Laser Scanning Images

As shown in Figure 4, before simulated oral processing, fat droplets were evenly dispersed in the gel matrix, and the coalescence of lipid droplets was limited. There was little difference between the samples. After simulated oral processing, the milk gel partially disintegrated and became loose. Although the fat droplets in all of the gel boluses showed partial coalescence and coalescence, the coalescence degrees of fat droplets in the gel boluses of EG20, EG40 and EG60 were greater when compared to the gel boluses of EG10 and EG85. Among them, EG40 showed the highest level of partial coalescence and coalescence of fat droplets distributed in the gel matrix. The results of confocal images are consistent with the results observed in particle size analysis.

For the emulsion system, when the emulsion enters the oral cavity, fat crystals inside fat droplets might penetrate the interface of fat droplets and bridge with another triglyceride molecule nearby, which leads to the coalescence of fat droplets [8]. At the same time, solid fat crystals might also melt at body temperature. When there are enough fat crystals in the interfacial layer, the time for solid fats to fully melt might be longer than that of partial coalescence, which could lead to an increase in the size of droplets [3]. For an emulsion-filled gel system, the fat droplets are entrapped in the biopolymer matrix. During oral processing, the gel matrix is first broken down by shearing and melting, followed by the release of fat droplets from the gel matrix and then coalescence. Therefore, the coalescence of fat droplets depends on the breakdown behavior of the gel matrix, the interaction between fat droplets and the properties of the droplets [5].

In the study [9], after shearing in a tribometer, emulsion-filled gels with high (35.7%) and medium (18.5%) SFC values yielded lower stability against coalescence than gels with low SFC (4.0%). Consistent with their study, negligible coalescence was found in acid milk gel with a low SFC in our study. Insufficient content of solid fat within fat droplets may allow fat droplets to melt before they are released from the gel matrix and thus make them less likely to coalesce. Furthermore, limited coalescence of fat droplets was also observed in acid milk gel with the highest SFC in our study. The reason may be that the droplets with high SFC have a dense network of crystals and thus lack sufficient liquid fat to connect with adjacent droplets [8]. Previous studies have shown that the optimal SFC for the partial coalescence of an emulsion is highly dependent on the content and composition of fat and the size, morphology and location of crystals [8,10]. In addition, the optimal SFC for fat droplets at rest may be lower than in the flow field, because when flow is applied, disruption may occur before the connections between the globules are formed [10]. In the present study, the optimal SFC in acid milk gel with 3% fat content to achieve coalescence during simulated oral processing was between 20% and 60%, with the highest at about 40%. The acid milk gels underwent gentle agitation during simulated oral processing, which may lead to an increase in the optimal SFC value of the acid milk gels. The coalescence of released fat droplets can exert a positive effect on the perception of creamy and fatty mouthfeel of emulsion-filled gels [1,5].

### 3.3. The Mechanical Properties of Acid Milk Gels during Simulated Oral Processing

#### 3.3.1. Apparent Viscosity

It has been widely reported that viscosity is strongly correlated with thickness perception, while thickness correlates positively with creaminess [30,31]. The apparent viscosity curves of acid milk gels with different SFCs as a function of the shear rate are shown in Figure 5. Before simulating oral processing, the apparent viscosity of acid milk gels decreased gradually with increasing shear rate (Figure 5A) due to shear thinning behavior [32]. The apparent viscosity of acid milk gels measured at a shear rate of 50/s tended to increase with increasing SFC (Table 3). This result is consistent with previous studies suggesting that higher SFC in droplets influences the rheological properties of gels by providing more densely packed oil droplets, as well as the anisometry and inhomogeneity of the droplet aggregates [9,24,33]. The difference between E10~E40 was not significant (*p* > 0.05), which may be due to the relatively low fat content of acid milk gels in the present study (3.0%), so the slight difference in SFC was not enough to affect the properties of the gel.

After simulated oral processing, the apparent viscosity of acid milk gels was much lower than that before oral processing, most likely due to the dilution effect of saliva and the increase in temperature. Furthermore, although the gel bolus with higher SFC in droplets exhibited higher viscosity at the initial stage of the apparent viscosity curve, the difference between samples narrowed with increasing shear rate (Figure 5B). There was no significant difference in apparent viscosity measured at a shear rate of 50/s between bolus gels (Table 3, *p* > 0.05). There may be two reasons for these results. On the one hand, mixing with saliva led to an order of magnitude decrease in the apparent viscosity of the gels, which may mask the differences between samples. On the other hand, the gradual melting of solid fat during simulated oral processing may also reduce the differences between samples.

#### 3.3.2. Tribological Properties

Tribology has been widely used to quantify the oral lubrication characteristics and creamy texture of emulsion systems [17,34,35]. As shown in Figure 5C, the friction curves of acid milk gels show two peaks, consistent with the previous observation [12]. At the beginning (sliding speed < 0.1 mm/s), the friction coefficients of all gel boluses significantly increased with the sliding speed, probably due to the stick-slip issue at the beginning of the test. With increasing sliding speed, the friction first decreased and then increased. The increasing regime is consistent with the mixed regime in the conventional Stribeck curve, probably due to the lubrication effect of free fat that is released from the gel matrix. It is worth noting that the mixed regime of E85 lasted much longer (sliding speed from 1 mm/s to about 30 mm/s) than other gel boluses. Most likely, the time needed for the SFC in gel bolus E85 to melt and coalesce was longer than in other gels, so the fat provided lubrication at higher sliding speeds. The subsequent decreasing regime is similar to the hydrodynamic regime in the traditional Stribeck curve. With the further development of the lubricating film in the contact zone, the internal friction controls the friction, which may cause the friction to increase linearly with the sliding speed [36,37]. Among them, the curve shape of the hydrophobic regime of gel bolus E40 was relatively flat, indicating that the lubrication film of gel bolus E40 was thinner in this regime, which was related to the highest coalescence level of fat droplets. The friction of the gel bolus decreased with the increase in sliding speed, probably due to the further breakdown of the protein gel matrix and the coalescence of the oil released from the matrix.

Regardless of sliding speed, gel bolus EG40 showed the lowest friction, while EG85 had the highest friction. With increasing SFC from 40% to 85%, friction nearly tripled at low speed (0.1–20 mm/s) and doubled at high speed (~100 mm/s). The order of friction coefficients at 20 mm/s was EG40 < EG20 ≤ EG60 < EG85 ≤ EG10 (Table 3), which is basically consistent with the coalescence results of fat droplets observed in the particle size distribution and confocal images (Figure 4). The coalescence of droplets can increase the oral lubricity by forming a smooth fat coating [1]. It should be mentioned that in order to have a specific coefficient of friction and a specific creaminess perception, a certain level of fat should accumulate on the surfaces of the oral tissues. This means that as the fat content increases, the required level of coalescence decreases, and the required SFC varies. Therefore, the optimal SFC to promote fat perception in emulsion-filled gels will vary with the fat content. In this study, acid milk gels with a fat content of about 3% with medium SFC (~40%) had the best lubricating properties because it had the highest level of fat coalescence.

### 3.4. Sensory Characterization

#### 3.4.1. QDA of Sensory Characterization

The QDA results of acid milk gels are shown in Table 4. Significant differences were found in the attributes of melting, mouth coating, smoothness and overall creaminess (*p* < 0.05), while there were no significant differences in graininess, spreadability or thickness between acid milk gels with different SFC values (*p* > 0.05). The ratings of smoothness and mouth coating in different acid milk gels were EG40 > EG20 ≥ EG60 > EG85 ≥ EG10, with a similar trend to the overall creaminess. This trend is consistent with the coalescence results of fat droplets observed in the confocal images and tribological properties (Figure 4 and Figure 5C).

In contrast to the previous study [9], our study showed for the first time that modulating the content of solid fat in an emulsion-filled gel can affect the perception of fat-related attributes of the gel. They found that gelatin gels with higher SFC values showed more coalescence and lower friction, but SFC did not affect the fat-related perceptions of the gels [9]. The discrepancy between our findings and the literature may be explained by the difference in the type of gel, the temperature of the prepared gel before oral processing and the fat content within the gel. The gel matrix was gelatin, which may cause the effect of SFC on the emulsion-filled gel to be masked. Additionally, the acid milk gels were kept at 4 °C in this study, while gelatin gels were kept at 20 °C before characterization in the study [9]. Therefore, in their study, after the gels entered the oral cavity (37 °C), the melting time of solid fat in the gels might be shorter than the time required for the gels to disintegrate and the fat droplets to partially coalesce. Furthermore, the relatively low fat content (3.0%) in our study may have amplified the effect of SFC differences. In the study [9], it was also found that for bound fat droplets with a fat content of 5%, the creamy mouthfeel intensities of gels containing high SFC were higher than those of the low-SFC group, although they were not significantly different at the 0.001 level.

#### 3.4.2. Temporal Dominance of Sensations Curves

As shown in Figure 6, except for mouth coating, all other sensory attributes basically showed a trend of first rising and then declining, while mouth coating showed a gradual upward trend. Furthermore, the dominance rates of the graininess of all samples were lower than the significance level and chance level, indicating that the graininess attribute of acid milk gel was not easily perceived in the oral cavity. A general overview of the TDS curves showed that melting and thickness were perceived as the first significantly dominant attributes, followed by spreadability, smoothness and mouth coating. For semi-solid foods such as acid milk gel, oral processing begins with movement between the tongue and the palate, inducing mechanical deformation and structural breakdown of the gel network, followed by surface attributes when rubbed against the palate. Thickness is known to be a bulk textural property perceived early during oral processing, while melting is a sensation that can be felt when fatty food experiences a temperature change as soon as it enters the oral cavity [38]. In the later stages of oral processing, the mouth coating is the dominant attribute, as this attribute is related to the thin layer covering the palate, which can be felt after swallowing. Similar results were observed in previous studies [39,40].

For acid milk gels with different SFCs, as the SFC increased, all of the rates of dominant attributes showed a trend of first rising and then declining. The maximum dominance rates of melting, smoothness and mouth coating were found in EG40. Furthermore, EG20 and EG40 both had the longest duration of melting dominance, while the longest duration of smoothness dominance was observed in EG40, consistent with the higher attribute intensities observed by QDA. For EG85, thickness was the first significantly dominant attribute, while melting had a lower dominance rate throughout the consumption, probably due to the higher onset melting temperature of fat, making fat-related attributes difficult to perceive. The TDS results complement the QDA results, showing that SFC in acid milk gels can affect the sequence and the duration of the dominant attributes, such as melting, smoothness and mouth coating of gels.

## 4. Conclusions

In this study, the relationship between the effect of SFC on the creaminess perception and the effect of SFC on the coalescence of fat droplets and oral tribological properties of acid milk gels was established. The results indicated that for acid milk gels with 3% fat content, the gel with an intermediate SFC (~40%) in the fat droplets had the greatest in-mouth coalescence level, the best lubricating properties, the highest creaminess rating and the longest melting and smoothness duration. In contrast, acid milk gels with too high or too low SFC had limited levels of in-mouth fat coalescence, higher friction coefficients and lower intensities of fat-related attributes. The study may aid in the understanding of the relationship between fat droplet characteristics and creaminess perception of emulsion gels and contribute to the development of low-fat foods with desirable sensory perception. Comparing the effects of SFC on emulsion-filled gels with different fat contents and different gelling agents is worthy of further study.

## Figures and Tables

**Figure 1 foods-11-02932-f001:**
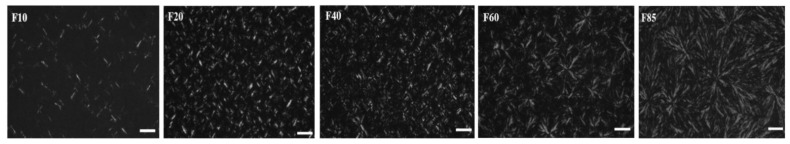
Polarized light micrographs of blended milk fats; scale bar 20 µm.

**Figure 2 foods-11-02932-f002:**
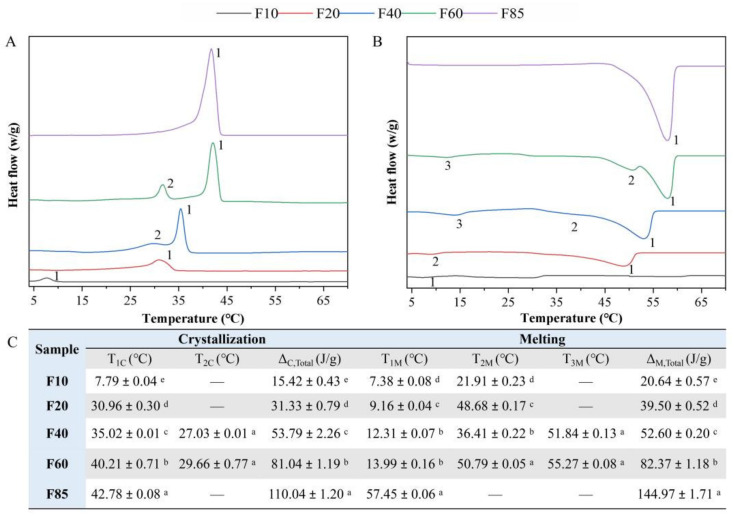
(**A**) Nonisothermal crystallization curves and (**B**) melting curves of blended milk fats; “1, 2, 3” indicate the first, second and third, respectively. (**C**) Nonisotermal crystallization melting temperature and total enthalpy of blended milk fats. T_1C_ and T_2C_ indicate the temperature of the first and second crystallization peaks; T_1M_, T_2M_ and T_3M_ indicate the temperature of the first, second and third melting peaks, respectively; Δ_C,Total_—the enthalpy of crystallization heat release; Δ_M,Total_—the enthalpy of heat absorbed by melting. “—” means not detected in the sample. Results are mean ± SD (*n* = 3). For each column, ^a–e^ mean that share the same letter within the same parameter were not significantly different (*p* ≥ 0.05).

**Figure 3 foods-11-02932-f003:**
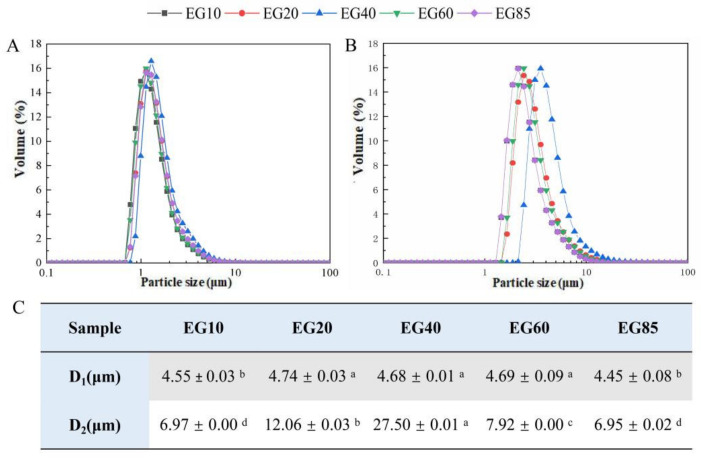
Particle size distribution of acid milk gels (**A**) before and (**B**) after simulated oral processing. (**C**) D_1_: volume diameter determined before simulated oral processing; D_2_: volume diameter determined after simulated oral processing. Results are mean ± SD (*n* = 3). For each row, ^a,b,c,d^ mean that share the same letter within the same parameter were not significantly different (*p* ≥ 0.05).

**Figure 4 foods-11-02932-f004:**
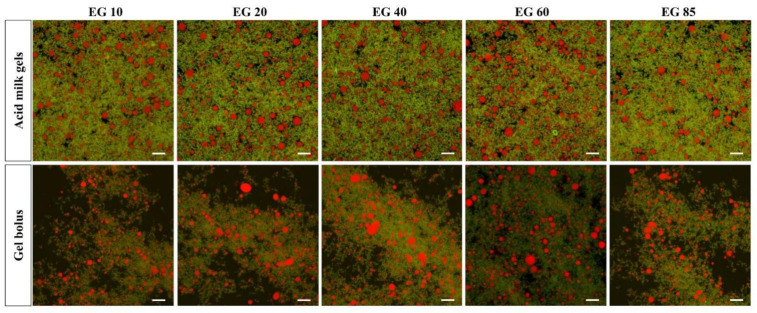
Microstructure of acid milk gels before and after simulated oral processing. Red: fat globules; green: protein; scale bar 15 μm.

**Figure 5 foods-11-02932-f005:**
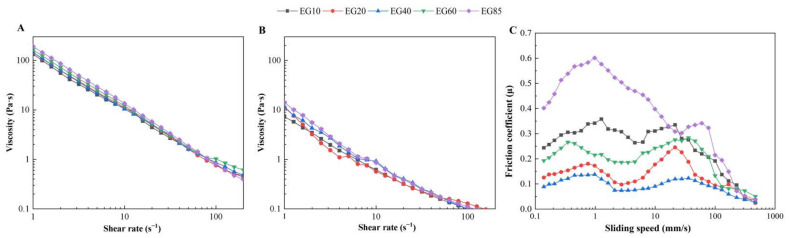
Rheological curves of acid milk gels (**A**) before and (**B**) after simulated oral processing. (**C**) Tribological curves of acid milk gels after simulated oral processing.

**Figure 6 foods-11-02932-f006:**
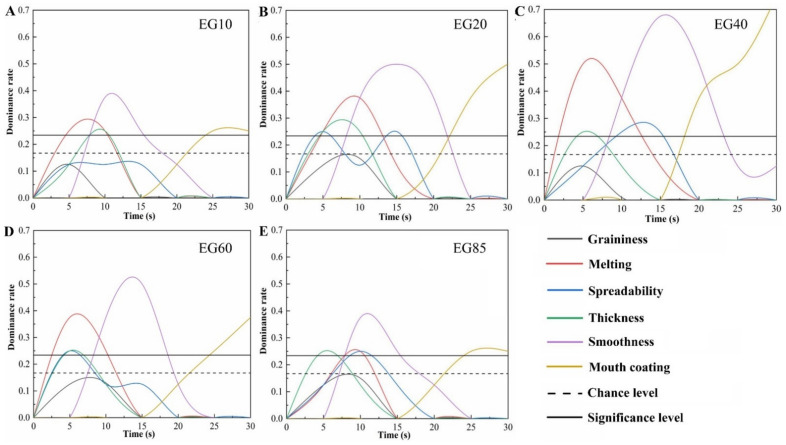
Temporal dominance of sensations curves of acid milk gels with different solid fat contents. (**A**–**E**) Acid milk gels: EG10, EG20, EG40, EG60 and EG85.

**Table 1 foods-11-02932-t001:** Formulas of blended milk fats.

Components	F10	F20	F40	F60	F85
Anhydrous milk fat (%)	30	30	50	50	10
Glyceryl stearate (%)	0	20	25	40	90
Oleic acid glyceride (%)	70	50	25	10	0

**Table 2 foods-11-02932-t002:** The sensory attribute list and evaluation criteria of quantitative descriptive analysis of acid milk gels.

Descriptor	Definition	Reference
Graininess	Dairy products containing grains, like particles in the mouth, which can be large and small.	Milk = 0; Oatmeal milk = 10
Melting	Dissolves in the mouth and the structure disappears, like ice cream melting.	Water = 0; Ice cream = 10
Spreadability	Unctuous, like chocolate spread.	Milk = 0; Chocolate sauce = 10
Thickness	Sample viscosity on the tongue. Sample is “flowing” if it flows immediately over the tongue. The sample is “sticky” if it stays on the tongue or flows slowly and is difficult to swallow.	Skim milk = 0; Cream = 3; Peanut butter = 10
Smoothness	The perception of lubricity or graininess when the sample is weakly compressed between the tongue and the palate and the tongue rubs repeatedly on the palate. The sample is “smooth” if it feels smooth and has no rough particles. The sample is “rough” if rough or irregular particles are felt in the mouth.	Peanut butter = 0; Skim milk = 3; Jelly pudding = 10
Mouth coating	The degree to which the coating is felt throughout the mouth; it can be felt on the teeth and upper jaw.	Whole milk = 5; Butter = 10
Overall creaminess	Cream is a soft, full feeling, and scores are based on a comprehensive feeling.	Skim milk = 0; Marshmallow sauce = 3; Cream = 10

**Table 3 foods-11-02932-t003:** Parameters of acid milk gels.

	EG10	EG20	EG40	EG60	EG85
Solid fat content	10.61 ± 0.41 ^a^	23.98 ± 0.86 ^b^	42.57 ± 0.12 ^c^	62.27 ± 0.38 ^d^	85.87 ± 0.13 ^e^
*η*^1^_50/s_ (Pa s)	1.65 ± 0.04 ^a^	1.60 ± 0.05 ^a^	1.61 ± 0.04 ^a^	1.74 ± 0.04 ^a,b^	1.86 ± 0.07 ^b^
*η*^2^_50/s_ (Pa s)	0.16 ± 0.02 ^a^	0.17 ± 0.01 ^a^	0.18 ± 0.02 ^a^	0.17 ± 0.01 ^a^	0.17 ± 0.01 ^a^
μ20	0.34 ± 0.07 ^b^	0.26 ± 0.01 ^b^	0.12 ± 0.01 ^a^	0.28 ± 0.02 ^b^	0.31 ± 0.04 ^b^

The data presented is the mean ± SD (*n* = 3). For each row, ^a–e^ mean that share the same letter within the same parameter were not significantly different (*p* ≥ 0.05). *η*^1^_50/s_—viscosity at a shear rate of 50/s before simulated oral processing; *η*^2^_50/s_—viscosity at a shear rate of 50/s after simulated oral processing; μ20—friction coefficient at a sliding speed of 20 mm/s after simulated oral processing.

**Table 4 foods-11-02932-t004:** Quantitative descriptive analysis evaluation scores of acid milk gels with different solid fat contents.

Descriptor	EG10	EG20	EG40	EG60	EG85
Graininess	3.25 ± 0.54 ^a^	3.13 ± 0.57 ^a^	3.35 ± 0.58 ^a^	3.55 ± 0.50 ^a^	3.65 ± 0.53 ^a^
Melting	3.42 ± 0.64 ^bc^	3.55 ± 0.50 ^bc^	4.00 ± 0.67 ^c^	3.00 ± 0.71 ^b^	2.06 ± 0.55 ^a^
Spreadability	5.07 ± 0.65 ^a^	5.56 ± 0.48 ^a^	5.60 ± 0.74 ^a^	5.38 ± 0.77 ^a^	4.88 ± 0.94 ^a^
Thickness	5.43 ± 0.55 ^a^	5.75 ± 0.48 ^a^	5.95 ± 0.60 ^a^	5.55 ± 0.60 ^a^	5.40 ± 0.52 ^a^
Smoothness	5.65 ± 0.53 ^a^	7.30 ± 0.59 ^b^	8.15 ± 0.47 ^c^	6.80 ± 0.63 ^b^	5.75 ± 0.59 ^a^
Mouth coating	5.95 ± 0.50 ^a^	6.70 ± 0.54 ^b^	7.65 ± 0.58 ^c^	6.30 ± 0.59 ^ab^	6.07 ± 0.72 ^a^
Overall creaminess	6.40 ± 0.48 ^a^	7.25 ± 0.54 ^b^	8.98 ± 0.54 ^c^	6.93 ± 0.69 ^ab^	6.53 ± 0.58 ^a^

Results are mean ± SD (*n* = 10). For each row, ^a,b,c^ mean that share the same letter within the same parameter were not significantly different (*p* ≥ 0.05).

## Data Availability

The data presented in this study are available on request from the corresponding author. The data are not publicly available.

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
