# Peer review of "Effect of Solid Fat Content in Fat Droplets on Creamy Mouthfeel of Acid Milk Gels"

_foods, 2022, doi:10.3390/foods11192932_

Round 1

Reviewer 1 Report

Authors presented a detailed work on Effect of solid fat content in fat droplets on creamy mouthfeel of acid milk gels. I have following observations on the submitted manuscript.

1.    What is the deviation percentage is expected while doing simulated oral processing than actual oral processing. Is there any study have done on this aspect???

2.    Line 162: What are those slight modifications? Is the modification on the RPM, temperature and time? If so, it required valid reasons for changing it. Because it will directly impact on the further studies on the characterization of gel bolus.

3.    Line 311/312: Check the statement for each column, different letters indicate significantly different values (P < 0.05) for Figure 3C??

4.    Please recheck the statistical analysis of Table 2.

5.    What happens if 3 % SFC is changed to higher or lower on the mouthfeel and creaminess??

Author Response

We sincerely appreciate your constructive comments. We have revised the manuscript according to your comments, and carefully proof-read the manuscript. Specific responses are as follows:

  1. What is the deviation percentage is expected while doing simulated oral processing than actual oral processing. Is there any study have done on this aspect???

  Answer: As mentioned by the reviewer, there are deviations between simulated oral processing and actual oral processing. Due to the complexity of the food system and the interactions between food matrices and oral conditions during oral processing, our understanding of the mechanisms underlying sensory perception is still limited. To assess sensory of foods often trained panels are used, while studies using rheological, tribological and structural methods to find the correlation of instrumental parameters with sensory attributes have also received much attention. The parameters of such tests were optimized to design test conditions closer to the physiological eating process, such as the simulating temperature, the inclusion of saliva and the selection of material that simulates the friction surface. Strong correlation between sensory data and instrument parameters performed under simulated oral processing have been verified in previous studies (Cheng, Liu, Shen, Liu, & Luo, 2020; Mu, Ren, Shen, Zhou, & Luo, 2022; Prinz, Janssen, & de Wijk, 2007; Stokes, Boehm, & Baier, 2013).

  1. Line 162: What are those slight modifications? Is the modification on the RPM, temperature and time? If so, it required valid reasons for changing it. Because it will directly impact on the further studies on the characterization of gel bolus.

  Answer: The modifications were the incorporation of human saliva and the difference in stirring time. In the study of Mu et al. (2021), the artificial saliva was used, and the artificial saliva was added to fermented milk at a ratio of 1:10 (w/w) and stirred manually at a shear rate around 50 rpm in a water bath at 37 °C for 15 s. In contrast, the human saliva were collected and used to better simulate the oral condition in our study. The acid milk gel samples were mixed with saliva at the same ratio and stirred at the same shear rate and temperature but for 20 seconds. According to the chewing result of acid milk gels by pre-experiments, 20 seconds can complete the oral processing process, while swallowing and then aftertaste could last about 10 seconds. Therefore, the simulated oral processing time was set for 20 s. The stirring time was set longer than the previous study because the weight and form of the samples were different. In the study of Mu et al. (2021), the sample was stirred fermented milk (more liquid) with a weight of 15 g. In our study, the sample was coagulated fermented milk (more solid) with a weight of 20 g. Consider the comment of the reviewer, the detailed information and the reason of the modification have been added to the text.

  1. Line 311/312: Check the statement for each column, different letters indicate significantly different values (P < 0.05) for Figure 3C??

  Answer: Thank you for your correction. The sentence has been revised into “For each row, different letters indicate significantly different values”.

  1. Please recheck the statistical analysis of Table 2.

  Answer: The statistical analysis has been redone in Table 2 (now Table 4).

  1. What happens if 3 % SFC is changed to higher or lower on the mouthfeel and creaminess??

  Answer: We deduce that the reviewer meant to say what happens if 3% fat content is changed to higher or lower on the mouthfeel and creaminess. As discussed in the manuscript, the optimum SFC for partial coalescence of emulsion is highly dependent on the content and composition of fat, and the size, morphology and location of crystals.

In order to have a specific coefficient of friction and a specific creaminess perception, a certain level of fat should accumulate on the surfaces of the oral tissues. This means that as the fat content increases, the required level of coalescence decreases, and the required SFC varies. However, for different gels with different fat contents, different SFC will exert different effects on sensory perception, which needs to be further clarified in follow-up studies.

  Thank you again for your valuable comments.

Reference

Cheng, W., Liu, H., Shen, Q., Liu, C., & Luo, J. (2020). A novel approach for modulating the spatial distribution of fat globules in acid milk gel and its effect on the perception of fat-related attributes. Food research international, 140(10), 109990.

Devezeaux de Lavergne, M., Strijbosch, V. M., Van den Broek, A. W., Van de Velde, F., & Stieger, M. (2016). Uncoupling the Impact of Fracture Properties and Composition on Sensory Perception of Emulsion‐Filled Gels. Journal of texture studies, 47(2), 92-111.

Laguna, L., Farrell, G., Bryant, M., Morina, A., & Sarkar, A. (2017). Relating rheology and tribology of commercial dairy colloids to sensory perception. Food & Function, 8(2), 563-573.

Mu, S., Liu, L., Liu, H., Shen, Q., & Luo, J. (2021). Characterization of the relationship between olfactory perception and the release of aroma compounds before and after simulated oral processing. Journal of dairy science, 104(3), 2855-2865.

Mu, S., Ren, F., Shen, Q., Zhou, H., & Luo, J. (2022). Creamy mouthfeel of emulsion-filled gels with different fat contents: Correlating tribo–rheology with sensory measurements. Food Hydrocolloids, 131, 107754.

Prinz, J. F., Janssen, A. M., & de Wijk, R. A. (2007). In vitro simulation of the oral processing of semi-solid foods. Food Hydrocolloids, 21(3), 397-401.

Stokes, J. R., Boehm, M. W., & Baier, S. K. (2013). Oral processing, texture and mouthfeel: From rheology to tribology and beyond. Current Opinion in Colloid & Interface Science, 18(4), 349-359.

Reviewer 2 Report

The manuscript titled “Effect of solid fat content in fat droplets on creamy mouthfeel of acid milk gels” is an original research article that investigated the effect of solid fat content in the fat droplets on the mouth feel of acid milk gels. The approach and results will interest readers and could provide good input for the foods industry. However, some thorough of English and incorporating the suggested corrections would improve the manuscript’s readability and help in meeting the journal publication standards.

Graphical abstract:

-          NA

Title:

-          The title is appropriate and explains the full scope of the study

Highlights:

-          NA

Abstract:

Line#11: However, effect – However, the effect

Line#14: needle-like but – needle-like, but

Line#14: vary-varies

Line#21: affect – affects

-          Line#10-11 explains the gap/need for this study, however an addition line of goal of the research, would add value to the abstract.

Keywords:

-          The selection of keywords sounds appropriate, however the first keyword – Solid fat content, could be changed to solid fat content

Other comments/corrections to improve the Manuscript:

Line#35: attentions – attention

Line#101: saliva were – saliva was

Line#105: in in – in

Line#132-134: please consider paraphrasing the sentence to improve the readability

Line#140: to fully dissolve – to dissolve fully

Line#161 shear rate around – shear rate of around

Line#168: was – were

Line#193: in in – in, please check for the correction throughout the manuscript

Line#200: rinse the mouth – rinse their mouth

Line#214: attract – attracted

Line#229: factors was – factor was/factors were

Line#237: are shown – is shown

Line#243: at shear rate – at a shear rate

Line#276: entalpy – enthalpy

Line#310: dimeter – diameter

Line#386: this results – these results

Line#414: with ever increase of – with the increase of

Line#454: can be also – can also be

Line#467: inducing in – inducing

Line#485: complements – complement

Author Response

Some thorough of English and incorporating the suggested corrections would improve the manuscript’s readability and help in meeting the journal publication standards.

Answer: We sincerely appreciate your constructive comments. We have revised the manuscript according to your comments, and the manuscript has been proofread by a native English speaker to correct the language problem. Specific responses are as follows:

Line#11: However, effect – However, the effect; Line#14: needle-like but – needle-like, but; Line#14: vary-varies; Line#21: affect – affects; Key word Solid fat content, could be changed to solid fat content; Line#35: attentions – attention; Line#101: saliva were – saliva was; Line#105: in in – in; Line#140: to fully dissolve – to dissolve fully; Line#161 shear rate around – shear rate of around; Line#168: was – were; Line#193: in in – in, please check for the correction throughout the manuscript; Line#200: rinse the mouth – rinse their mouth; Line#214: attract – attracted; Line#229: factors was – factor was/factors were; Line#237: are shown – is shown; Line#243: at shear rate – at a shear rate; Line#276: entalpy – enthalpy; Line#310: dimeter – diameter; Line#386: this results – these results; Line#414: with ever increase of – with the increase of; Line#454: can be also – can also be; Line#467: inducing in – inducing; Line#485: complements – complement.

Answer: As the reviewer suggested, these words have been revised.

-          Line#10-11 explains the gap/need for this study, however an addition line of goal of the research, would add value to the abstract.

Answer: As the reviewer suggested, the aim of this research has been added to the abstract.

Line#132-134: please consider paraphrasing the sentence to improve the readability

Answer: As the reviewer suggested, the sentence has been rephrased.

 Thank you again for your valuable comments.

Reviewer 3 Report

The main objective of this study was to determine the effect of SFC in fat droplets on the creamy mouthfeel of acid milk gel. The contribution of fat to the sensory properties of foods is extremely important, as is understanding how fat affects the sensory properties of foods. The results of the research presented in the manuscript can undoubtedly help to understand the relationship between fat droplet characteristics and the perception of the creaminess of emulsion gels. Such knowledge can also contribute to the development of low-fat foods with desirable sensory perception. The obtained results are in accordance with the topic of the Special Issue of Foods Journal, "Multi-Sensory Appreciation and Evaluation toward Foods: Gustatory, Sight, Touch, Smell". However, several points need to be considered:

L106: I would suggest placing Table S1 directly in the manuscript's text. This will make the manuscript more readable.

L147-148: How long did the pH reduction process take? How did it trend in each sample? Were differences found in the time taken to obtain the gels?

L162: What did the modifications include? I think it is worth describing the indicated modifications in research methods.

L190-202: Please indicate the source of literature.

L193: I would suggest that Table S2 could be placed in the text of the manuscript. Like in line 106 (Table S1), it will make the manuscript more readable.

L240: Letters denoting significant differences should be written as superscripts next to the average value. It is assumed that letter designations are placed from the lowest average value to the highest. Please review the other tables and follow my comment. 

 Was no statistical analysis done for the parameter "η250/s"? If the average values were not significantly different, denote them with the same letter.

Author Response

We sincerely appreciate your constructive comments. We have revised the manuscript according to your comments, and carefully proof-read the manuscript. Specific responses are as follows:

L106: I would suggest placing Table S1 directly in the manuscript's text. This will make the manuscript more readable.

Answer: As the reviewer suggested, the Table S1 has been placed in the manuscript as Table 1.

L147-148: How long did the pH reduction process take? How did it trend in each sample? Were differences found in the time taken to obtain the gels?

Answer: The pH reduction process took about 4 hours (3.95±0.10 h). There was no significant difference observed among these gels.

L162: What did the modifications include? I think it is worth describing the indicated modifications in research methods.

Answer: The modifications were the incorporation of human saliva and the difference in stirring time. In the study of Mu et al. (2021), the artificial saliva was used, and the artificial saliva was added to fermented milk at a ratio of 1:10 (w/w) and stirred manually at a shear rate around 50 rpm in a water bath at 37 °C for 15 s. In contrast, the human saliva were collected and used to better simulate the oral condition in our study. The acid milk gel samples were mixed with saliva at the same ratio and stirred at the same shear rate and temperature but for 20 seconds. According to the chewing result of acid milk gels by pre-experiments, 20 seconds can complete the oral processing process, while swallowing and then aftertaste could last about 10 seconds. Therefore, the simulated oral processing time was set for 20 s. The stirring time was set longer than the previous study because the weight and form of the samples were different. In the study of Mu et al. (2021), the sample was stirred fermented milk (more liquid) with a weight of 15 g. In our study, the sample was coagulated fermented milk (more solid) with a weight of 20 g. Consider the comment of the reviewer, the detailed information and the reason of the modification have been added to the text.

L190-202: Please indicate the source of literature.

Answer: As the reviewer suggested, the reference has been added to the manuscript.

L193: I would suggest that Table S2 could be placed in the text of the manuscript. Like in line 106 (Table S1), it will make the manuscript more readable.

Answer: As the reviewer suggested, the Table S2 has been placed in the manuscript as Table 2.

L240: Letters denoting significant differences should be written as superscripts next to the average value. It is assumed that letter designations are placed from the lowest average value to the highest. Please review the other tables and follow my comment. 

Answer: The letters in all tables have been revised as suggested by the reviewer.

Was no statistical analysis done for the parameter "η250/s"? If the average values were not significantly different, denote them with the same letter.

Answer: There was no significantly difference in η250/s between samples. The letter has been added as suggested by the reviewer.

Thank you again for your valuable comments.

Reference

Cheng, W., Liu, H., Shen, Q., Liu, C., & Luo, J. (2020). A novel approach for modulating the spatial distribution of fat globules in acid milk gel and its effect on the perception of fat-related attributes. Food research international, 140(10), 109990.

Devezeaux de Lavergne, M., Strijbosch, V. M., Van den Broek, A. W., Van de Velde, F., & Stieger, M. (2016). Uncoupling the Impact of Fracture Properties and Composition on Sensory Perception of Emulsion‐Filled Gels. Journal of texture studies, 47(2), 92-111.

Laguna, L., Farrell, G., Bryant, M., Morina, A., & Sarkar, A. (2017). Relating rheology and tribology of commercial dairy colloids to sensory perception. Food & Function, 8(2), 563-573.

Mu, S., Liu, L., Liu, H., Shen, Q., & Luo, J. (2021). Characterization of the relationship between olfactory perception and the release of aroma compounds before and after simulated oral processing. Journal of dairy science, 104(3), 2855-2865.

Mu, S., Ren, F., Shen, Q., Zhou, H., & Luo, J. (2022). Creamy mouthfeel of emulsion-filled gels with different fat contents: Correlating tribo–rheology with sensory measurements. Food Hydrocolloids, 131, 107754.

Prinz, J. F., Janssen, A. M., & de Wijk, R. A. (2007). In vitro simulation of the oral processing of semi-solid foods. Food Hydrocolloids, 21(3), 397-401.

Stokes, J. R., Boehm, M. W., & Baier, S. K. (2013). Oral processing, texture and mouthfeel: From rheology to tribology and beyond. Current Opinion in Colloid & Interface Science, 18(4), 349-359.